# BEYOND POLICY TRAINING: SOLUTION SEARCH VIA TASK FLOW LEARNING AND PLANNING FROM UNLABELED VIDEOS

## ABSTRACT

Traditional policy learning in reinforcement learning relies on costly annotated data from extensive environment interaction. In contrast, massive unlabeled videos contain rich task knowledge but remain underutilized. Inspired by how humans acquire skills from watching videos, we propose Policy-Free Flow Search (PFFS). Not depending on explicit policies, PFFS learns to understand tasks through temporal consistency in single demonstrations and structural alignment across them. It models task stage transitions autoregressively to form a coherent task flow. At deployment, PFFS performs backward planning to generate a goal-to-initial task flow, then executes forward search to solve the task along this flow with minimal exploration. For further utility, we extend PFFS to PFFS-RL, an reinforcement learning (RL) framework using save-point-structured trajectories and task-flow-aligned rewards, significantly boosting exploration efficiency. Experiments show PFFS solves Minecraft tasks with very few exploration in a policy-free manner, while PFFS-RL outperforms other RL baselines with improved exploration under the same data volume. This work introduces a novel policy-free paradigm to leverage unlabeled videos for efficient task solving, advancing decision-making in resource-constrained scenarios.

## 1 INTRODUCTION

Reinforcement learning (RL) has demonstrated remarkable capabilities in solving complex decision-making tasks by learning policies that map states to optimal actions (Kaelbling et al., 1996). However, the RL paradigm faces a critical challenge in practical deployment: policy training depends on sufficient high-quality data (Ma et al., 2025a; Ball et al., 2023; Ma et al., 2024). The acquisition of such data typically requires either extensive interaction with the environment or manual annotation by human experts. The former suffers from poor quality while the latter is costly. The data's poor generalization (Karbasi et al., 2023) presents a further challenge. Bound to the agent's exact state-action mappings, the data collected is sensitive to noise and hardly reusable across embodiments or environments. Given these challenges, acquiring low-cost, high-quality data is a fundamental problem in RL.

To address this problem, a major direction focuses on efficient exploration with prior knowledge (Ladosz et al., 2022). Current methods either rely on handcrafted prior cues (Liang et al., 2022) or leverage annotated data to learn exploration guidance (Baker et al., 2022), which still suffers from annotation costs. More critically, the task-solving knowledge, encoded as state-action mappings within agent-specific, noise-sensitive policies, prevents generalization across embodiments or environments (Ghosh et al., 2021). However, a vast source of embodiment-agnostic data exists in practice, i.e., unlabeled videos. Such videos are abundant, low-cost, and inherently contain rich task-solving knowledge that existing RL methods lack the ability to effectively extract and leverage. This gap motivates our work and raises a key question:

*How to extract generalizable solutions from unlabeled videos?*

In response to this question, we propose a functional prototype. This prototype is a parameter-free, logic-driven blueprint that intuitively illustrates our motivation. It does not rely on explicit,

agent-specific policy modeling, but extracts structured task flow to guide efficient forward search. As shown in Fig.1, the prototype works in two stages. First, backward planning starts from the goal state $s_{\text{goal}}$ and iteratively predicts a sequence of one-step forward reachable sets $seq(\mathcal{R}) = (\mathcal{R}_0 = \{s_{\text{goal}}\}, \mathcal{R}_{-1}, \cdots, \mathcal{R}_{-N})$ in reverse order, until the start state $s_{\text{start}} \in \mathcal{R}_{-N}$. Second, forward searching uses the backward planned sets sequence $seq(\mathcal{R})$ to guide exploration from $s_{\text{start}}$ to $s_{\text{goal}}$, ensuring an on-track trajectory by following $seq(\mathcal{R})$.

The prototype relies only on state sequences, a property aligning naturally with unlabeled video demonstrations, which capture task-relevant state changes without action annotations. We instantiate the prototype by modeling the reachable sets $\mathcal{R}_{-k}$ as task flow categories extracted from these videos, yielding Policy-Free Flow Search (PFFS). Specifically, PFFS self-supervises task flow encoding autoregressively through two consistency constraints: in-flow consistency ensures coherent temporal progression within single videos, while cross-flow consistency aligns structural logic across multiple videos. For broader RL applicability, we combine PFFS trajectory search with RL, yielding PFFS-RL. Experiments show that by pretraining on unlabeled video demonstrations, PFFS can find solutions with very few steps of environmental exploration. Notably, PFFS-RL outperforms baseline RL algorithms even without external reward signals.

The key contributions of this paper are as follows: **1**. We present a novel method to effectively leverage unlabeled video demonstrations for task-solving, which first extracts task solution knowledge from video-captured state transitions and then encodes such knowledge into generalizable task flows. **2**. We design PFFS, a policy-free framework that achieves efficient and generalizable solution exploration via the extracted task flows, avoiding agent-specific policy modeling.

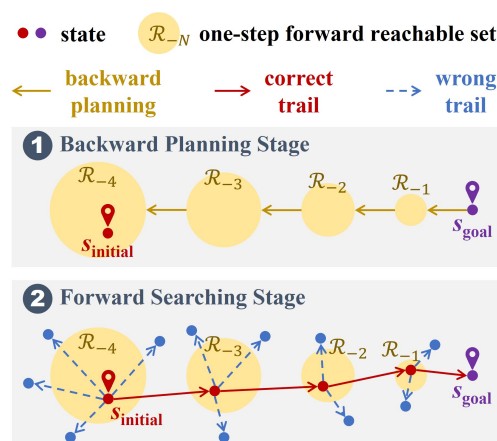

Figure 1: Foundational Prototype of PFFS. In backward planning, a set sequence $seq(\mathcal{R}) = (\mathcal{R}_0 = \{s_{\text{goal}}\}, \mathcal{R}_{-1}, \cdots, \mathcal{R}_{-N})$ is acquired via iteratively backward prediction. $\mathcal{R}_{-k}$ ($k \in [0, N]$) denotes the set of states reachable in one step from states in $\mathcal{R}_{-(k-1)}$. Forward searching uses $seq(\mathcal{R})$ to guide exploration from $s_{\text{start}}$ to $s_{\text{goal}}$. Wrong trails (dashed blue lines) deviate from $seq(\mathcal{R})$, while the correct trails (solid red lines) follow it. Since $\forall s_{-(k-1)} \in \mathcal{R}_{-(k-1)}$ is reachable from some state in $\mathcal{R}_{-k}$, we are guaranteed to iteratively find an on-track trajectory from $\mathcal{R}_{-N}$ (containing $s_{\text{start}}$) to $\mathcal{R}_0 = \{s_{\text{goal}}\}$.

**3**. We extend PFFS to PFFS-RL, which boosts RL exploration efficiency via task flow guided exploration and outperforms baseline RL algorithms even without external rewards.

## 2 BACKGROUNDS

### 2.1 REINFORCEMENT LEARNING

Reinforcement Learning (RL) is a framework for sequential decision-making, where an agent optimizes behavior through interaction with external environments (Sutton et al., 1999). RL is formally modeled as a Markov Decision Process (MDP) $(\mathcal{S}, \mathcal{A}, p, r, \gamma)$. $\mathcal{S}$ is the state space, containing all possible environmental configurations $s \in \mathcal{S}$. $\mathcal{A}$ is the action space, containing all actions $a \in \mathcal{A}$ available to the agent. $p : \mathcal{S} \times \mathcal{A} \times \mathcal{S} \to [0, 1]$ is transition probability of moving from $s$ to $s'$ via action $a$. $r : \mathcal{S} \times \mathcal{A} \to \mathbb{R}$ is reward function quantifying action desirability in state $s$. $\gamma \in [0, 1)$ is discount factor weighting future rewards.

The agent learns a policy $\pi : \mathcal{S} \to \Delta(\mathcal{A})$ (probability distribution over actions) to maximize the expected cumulative discounted reward: $J(\pi) = \mathbb{E}_{\pi,p} \left[ \sum_{t=0}^{\infty} \gamma^t r(s_t, a_t) \mid s_0 \sim \rho_0 \right]$, where $s_t, a_t$ are the state and action at step $t$, and $\rho_0$ is the initial state distribution. RL algorithms optimize $J(\pi)$ via value function estimation (e.g., $V^\pi(s) = \mathbb{E}_{\pi,p} \left[ \sum_{k=0}^{\infty} \gamma^k r(s_{t+k}, a_{t+k}) \mid s_t = s \right]$) or direct policy updates, using interaction-collected tuples $(s, a, r, s')$. A critical limitation of RL in practical

deployment lies in: policy optimization relies on large volumes of high-quality interaction data, as learning from limited experience often leads to suboptimal or unstable policies.

## 2.2 Efficient Exploration in RL

Exploration in Reinforcement Learning (RL) refers to the process where an agent interacts with the environment through trial and error to acquire experience data. The quality and efficiency of exploration directly determine how quickly and effectively an agent can learn to solve complex tasks (Ladosz et al., 2022). The most widely used exploration strategy is the $\epsilon$-greedy (Dann et al., 2022), which randomly selects actions with a fixed probability to balance exploration of uncharted regions and exploitation of known rewards. Beyond this simple heuristic, two major categories of exploration strategies are prevalent: intrinsic motivation-based methods (Aubret et al., 2023; Colas et al., 2022) and uncertainty-based methods (Zangirolami & Borrotti, 2024; Lockwood & Si, 2022). The former incentivize exploration by designing internal reward signals to stimulate its curiosity about unvisited parts of the environment. By contrast, the latter guide exploration toward regions where the agent's knowledge is most limited.

Recently, methods that leverage prior knowledge to structure exploration have gained growing attention (Ma et al., 2025b; Metzger et al., 2024). The core idea of these methods is using learned internal knowledge or handcrafted external priors as guiding signals, avoiding blind trial-and-error and leading the agent to explore more valuable regions. For instance, the Prioritized Trajectory Guided Model (PTGM) (Yuan et al., 2024) learns trajectory priors via a pre-trained goal prior model (on clustered goal states), which is used to guide its high-level policy in prioritizing high-potential path exploration. Video PreTraining (VPT) (Baker et al., 2022) trains an inverse dynamic model on labeled data to annotate actions for unlabeled videos, which are then used for pre-training to enhance exploration efficiency. However, these methods are still limited to agent-specific, task-bound state-action mappings, which constitute low-level and noise-sensitive information. This narrow focus results in the neglect of the high-level, generalizable task-solving knowledge in unlabeled videos.

## 3 Methodology

### 3.1 Overview of Policy Free Flow Search Framework

In this section, we introduce Policy-Free Flow Search (PFFS), an agent-agnostic, policy-independent task guidance framework leveraging unlabeled video demonstrations. PFFS aims to extract structured task-solving knowledge from video frames and transform it into abstract task flows, thereby providing universal task-solving guidance across diverse agents. This design addresses two key limitations of traditional reinforcement learning (RL): heavy reliance on costly high-quality annotated data, and strong coupling between learned policies and an agent's specific embodiment. PFFS operates in two stages: offline learning and online deployment. This subsection focuses on the former. Details of online deployment are provided in Section 3.3.

The offline training pipeline of PFFS is illustrated in Fig.2. First, PFFS starts with a dataset of unlabeled video demonstrations $D = V_1, V_2, \cdots, V_N$, where video $V_i$ ($i \in [1, N]$) consists of frames $V_i = \{s_{1,i}, s_{2,i}, \cdots, s_{T_i,i}\}$. $T_i$ is the frame number. $V_i$ is processed by a VQVAE (Vector Quantized AutoEncoder) (Van Den Oord et al., 2017) composed of an encoder (Enc) and a decoder (Dec). The encoder maps frame $s_{t,i}$ ($t_i \in [1, T_i]$) into latent vector $z_{t,i}$, while the decoder reconstructs the $s_{t,i}$ from $z_{t,i}$ to ensure visual fidelity. After obtaining latent vectors for all frames across $D$, we apply DBSCAN (Schubert et al., 2017) to cluster these latent vectors, yielding a set of cluster centers $C = \{C_1, C_2, \cdots, C_K\}$, where $K$ is the number of clusters. Each cluster center $C_k$ corresponds to a category $c_k$. Each latent vector $z$ is then assigned to a category. For video $V_i$, the sequence of its frames' categories *is defined as the task flow* of $V_i$, denoted as $TF_i = \{c_{1,i}, c_{2,i}, \cdots, c_{T_i,i}\}$, where $c_{t,i}$ refers to the category ssigned to the latent vector $z_{t,i}$ of frame $s_{t,i}$.

We design two consistency constraints to refine the VQVAE's latent representations. The first is in-flow consistency, which enforces monotonic and uniform progression of latent vectors. The second is cross-flow consistency, which leverages an autoregressive backward planner (BP) to align task flow structures across different video demonstrations, ensuring generalizability beyond single demonstrations. Next we focus on extracting task-specific temporal logic from these latent representations via in-flow and cross-flow consistencies.

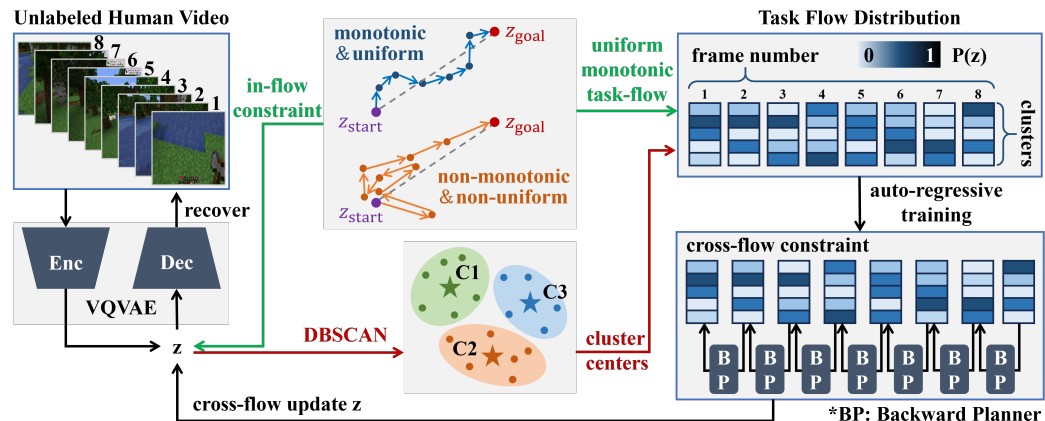

Figure 2: Offline training pipeline of Policy-Free Flow Search (PFFS). PFFS takes unlabeled video frames as input. these frames are processed through a VQVAE with frame reconstruction and in-flow monotonicity constraints, DBSCAN clustering to identify latent centers ($C1, C2, C3$), and an autoregressive backward planner (BP) to model the transition of task flow distributions. The pipeline iteratively refines latent vectors z by integrating visual, in-flow, and cross-flow constraints.

## 3.2 TASK-FLOW CONSISTENCY CONSTRAINTS

**In-Flow Consistency**. A single video demonstration inherently captures the temporal logic of task execution that form a coherent task process. We refer to this sequential logic within an individual video as the in-flow. For latent representations $\{z_1, z_2, \cdots, z_T\}$ to effectively capture such logic, they should satisfy two core requirements: reflecting the directional progression of the task and maintaining uniform progression of latent changes. To meet these requirements, we propose the in-flow consistency constraint, implemented via two complementary loss functions.

*First*, we define the global task direction vector using the initial and final frame latents: $\text{flow}_{\text{vec}} = z_T - z_1$. To enforce the alignment between this global direction and latent transition $z_{j+1} - z_j$ ($j \in [1, T-1]$), we introduce the directional consistency loss via cosine similarity:

$$\mathcal{L}_{\text{dire}} = \frac{1}{T-1} \sum_{j=1}^{T-1} \left(1 - \cos\left(z_{j+1} - z_j, \text{flow}_{\text{vec}}\right)\right) \tag{1}$$

Minimizing this loss enforces all latent transitions contribute positively to the global task direction, avoiding reversed or oscillating steps. *Second*, for steady latent progression, we compute the projection of latent transition $z_{j+1} - z_j$ on the directional vector $\text{flow}_{\text{vec}}$, denoted by $l_j = (z_{j+1} - z_j) \cdot \frac{\text{flow}_{\text{vec}}}{\|\text{flow}_{\text{vec}}\|}$. $l_j$ quantifies the stepwise progress along the global task direction. For uniform progression, we introduce the uniform loss:

$$\mathcal{L}_{\text{uni}} = \frac{1}{T-1} \sum_{j=1}^{T-1} \left(l_j - \frac{|\text{flow}_{\text{vec}}|}{T-1}\right)^2 \tag{2}$$

Minimizing this loss ensures each progress step is close to the ideal uniform value, preventing extreme clustering or abrupt jumps in the latent space.

**Cross-Flow Consistency**. Multiple video demonstrations $\{V_1, V_2, \cdots, V_N\}$ of the same task share task flows $\{TF_1, TF_2, \cdots, TF_N\}$ with consistent structural progression. We refer to this consistency of task flow structures across demonstrations as cross-flow consistency, which ensures uniform and generalizable task flow modeling for the same task.

To meet this requirement, we propose the cross-flow consistency constraint, implemented via an auto-regressive loss. First, we identify cluster centers $\{C_1, C_2, \cdots, C_K\}$ using DBSCAN across the latents of unlabeled video set $D$. For each latent $z_{t,i}$, we compute its distance to each cluster center $C_k$, denoted by $d_{k,t,i}$. Then we derive the category probability distribution:

$$P(c_k|z_{t,i}) = \text{Softmax}\left(-d_{1,t,i}, -d_{2,t,i}, \cdots, -d_{K,t,i}\right) \tag{3}$$

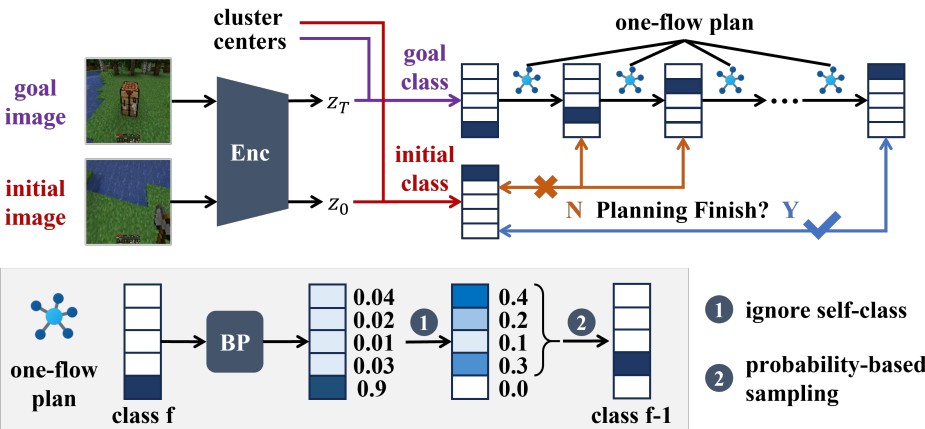

Figure 3: Backward planning pipeline of PFFS. Initial and goal images are first encoded into latent vectors $z_0$ and $z_T$. These vectors are then mapped to their respective flow categories. A backward planner iteratively executes one-flow plan steps: in each step, it samples the prior flow category from the probability with self-category ignored. This iterative process continues until a complete task flow plan from goal to initial state is generated.

where $P(c_k|z_{t,i})$ denote the probability of $z_{t,i}$ belonging to category $c_k$, and $t_{k \to m}$ denote the transition probability from category $k$ to $m$ learned by the planner. The cross-flow loss enforces auto-regressive consistency by penalizing the deviation between predicted and actual task flow probabilities through the minimization of a Kullback-Leibler (KL) divergence:

$$\mathcal{L}_{\text{cross-flow}} = \sum_{i=1}^{N} \sum_{t=2}^{T_i} \sum_{k=1}^{K} P(c_k|z_{t-1,i}) \log \frac{P(c_k|z_{t-1,i})}{\sum_{m=1}^{K} P(c_m|z_{t,i}) \cdot t_{m \to k}} \tag{4}$$

Minimizing this loss ensures that the latent clusters form a consistent, auto-regressive task flow structure across all video demonstrations, enabling the backward planner (BP) to learn generalizable task-solving guidance.

**Total Training Loss**. The total training loss of PFFS is the weighted sum of multiple components to balance visual fidelity, single-video temporal logic, and cross-video structural consistency:

$$\mathcal{L}_{\text{total}} = \mathcal{L}_{\text{rec}} + \alpha \cdot \mathcal{L}_{\text{dire}} + \beta \cdot \mathcal{L}_{\text{uni}} + \gamma \cdot \mathcal{L}_{\text{cross-flow}} \tag{5}$$

where $\alpha, \beta, \gamma > 0$, $\mathcal{L}_{\text{rec}}$ is the reconstruction loss of VQVAE ensuring frame visual fidelity, $\mathcal{L}_{\text{dire}}$ and $\mathcal{L}_{\text{uni}}$ form the in-flow consistency loss.

### 3.3 DEPLOYMENT AND EXTENSION

To illustrate the deployment process, we detail backward planning and forward searching as follows.

**Backward Planning**. Fig.3 illustrates the backward planning process. First, the initial image $s_{init}$ and goal image $s_{goal}$ are encoded into latent vectors $z_{init}$ and $z_{goal}$ using the encoder (Enc) of VQ-VAE. $z_{init}$ and $z_{goal}$ are then mapped to initial task flow category $c_{init}$ and goal task flow category $c_{goal}$ using the precomputed cluster centers from DBSCAN. The backward planner (BP) generates task flow plan $\{c_{goal}, \cdots, c_{init}\}$ by iteratively executes an one-flow plan (plan to last category). For a given flow category $k$ ($k \in [1, K]$), BP outputs the probability distribution over all possible prior flow categories; we ignore the self-category and samples the prior flow category based on the probability distribution. This process repeats until the initial flow category $c_{init}$ is reached, forming a sequential plan from the goal to the initial state.

**Forward Searching**. Fig.4 illustrates the forward searching process. The agent follows the planned task flow $TF = \{c_{goal}, \cdots, c_{init}\}$ from the initial category $c_{init}$ to the goal category $c_{goal}$. In each one-flow search (search to next category), the agent executes random actions step by step, computes the task flow category of the current state, and checks alignment with the planned sequence. When

encountering a new category that matches the planned flow for the first time, this state is recorded as a save point. If the agent enters a category that deviates from the plan or reaches the maximum search step length, it returns to the latest save point or $s_{init}$ if no save points exist. This process repeats until the $s_{goal}$ is reached. Save points enable the agent to resume exploration from validated stages, avoiding redundant re-exploration of early steps.

**Extension to PFFS-RL.** We further extend PFFS to reinforcement learning (PFFS-RL) to boost exploration efficiency. PFFS-RL retains the backward planning and forward search mechanisms and integrates a task-flow-aligned reward signal to guide policy learning. Specifically, we define a structured reward: $r$ is *positive* when encountering a new category that matches the planned flow for the first time, i.e., reaching a new save point; $r$ is *negative* for entering an off-track category; and a small *step penalty* is applied to avoid unproductive exploration. An example of the reward design is provided in Fig.4.

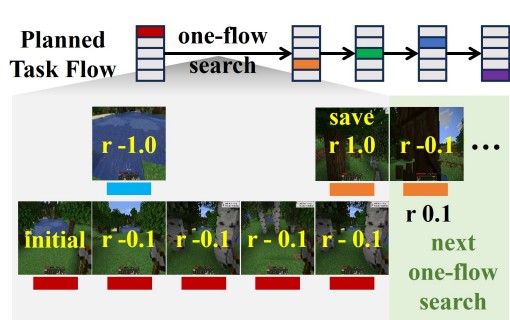

Figure 4: Forward Searching in PFFS. The agent follows the planned task flow generated by backward planning and executes one-flow search iteratively. Each search begins with latest save point, while terminates once reaching the maximum step length or the next flow category. The first state of a new on-track flow category is recorded as a save point. We also show the reward design for PFFS-RL training.

For sample collection, PFFS-RL structures trajectories by leveraging save points. Each exploration episode starts from the latest save point and runs until either the next save point is reached or the episode terminates due to off-track deviation or exceeding the maximum search length. The trajectory segment from the save point to termination is recorded as $seg_{post}$, and the pre-save-point trajectory (from the initial state to the save point) is stored as $seg_{save}$. These segments are concatenated as $(seg_{save}, seg_{post})$ to form complete training samples. This approach reuses validated early-stage trajectories while focusing exploration on uncharted segments, significantly reducing redundant data collection compared to vanilla RL.

## 4 EXPERIMENTS

### 4.1 EXPERIMENTAL SETUP

We chose Minecraft (Fan et al., 2022) as our benchmark, a widely used open-world sandbox game that serves as a pivotal platform for evaluating reinforcement learning (RL) methods on long-horizon tasks and exploration efficiency. Its unstructured environment and sequential task dependencies make it well-suited for validating our policy-free, video-guided framework. Our experiments leverage the OpenAI Contractor Dataset (Baker et al., 2022), from which we use 1200 video clips covering five representative downstream tasks: Harvest water with bucket, Harvest log in plains, Mine cobblestone, Mine iron ore, and Harvest sand. To establish an initial understanding of task progression, we first warm up the VQVAE and DBSCAN on a small, manually annotated subset of 50 clips. This step allows the model to learn a mapping from video frames to distinct task stages, which is essential for guiding the agent without engineered rewards.

Downstream RL tasks are conducted in the MineDojo simulator (Fan et al., 2022) using Proximal Policy Optimization (PPO) (Schulman et al., 2017). The agent's learning is guided by a novel, task-flow-aligned intrinsic reward: it receives +1.0 for actions that contribute to entering the next task flow stage and -1.0 for deviating from the current stage. This structure allows the agent to learn complex behaviors directly from unlabeled video. We compare our method, PFFS-RL, against four baselines: Director (PPO), PPO with CLIP Reward (Fan et al., 2022), PTGM (Yuan et al., 2024), and VPT (Baker et al., 2022), Steve-1 (Lifshitz et al., 2023). All baselines are trained with the same data volume, but they rely on the dataset's native dense extrinsic rewards and use conventional trajectory acquisition methods. In contrast, our method derives guidance solely from unlabeled video. Performance is evaluated by comparing task success rates and the average steps to completion. More implementation details are provided in Appendix B.

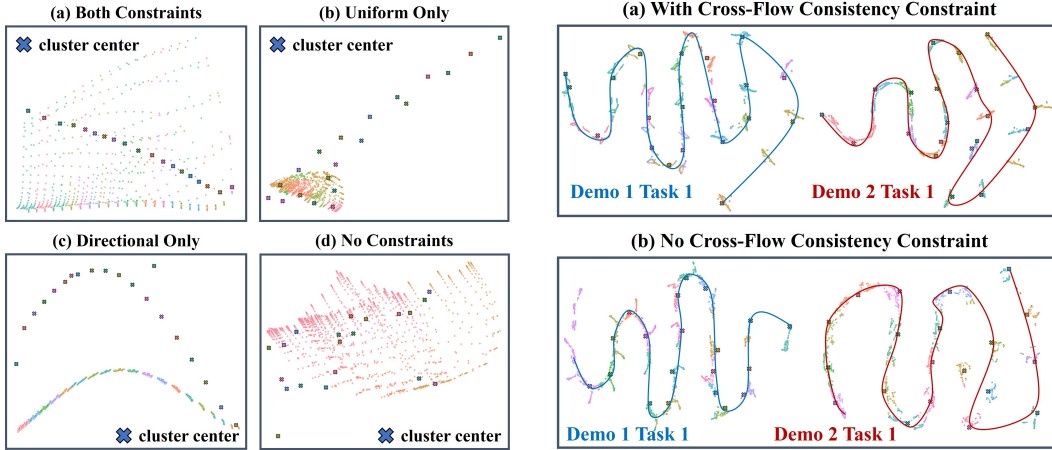

Figure 5: Visualization of in-flow constraint ablation. The figure displays 2D PCA projections of learned latent spaces under various in-flow consistency constraints. (a) Both directional and uniform constraints produce ordered, linear trajectories. (b) Only the uniform constraint results in fragmented, chaotic paths. (c) Only the directional constraint creates a non-linear, uneven arc. (d) Without any constraints, the latent space is unstructured and scattered.

Figure 6: Visualization of cross-flow constraint ablation. This figure displays 2D t-SNE projections of learned latent spaces with and without cross-flow constraints. The cluster centers are connected by solid lines to show the manifold structure. (a) With the cross-flow constraint, the two trajectories express similar manifolds. (b) Without the cross-flow constraint, the two trajectories remain inconsistent, failing to form a unified representation.

## 4.2 Evaluation of In-Flow and Cross-Flow Constraints

To validate the effectiveness of the in-flow and cross-flow consistency constraints in shaping task-relevant latent representations, we conduct two experiments: (1) the impact of in-flow constraints on latent space structure; (2) the generalization of cross-flow constraints across demonstrations. Demonstrations of learned task flows are presented in Appendix A.

**In-Flow Constraint Evaluation**. We conducted an ablation study to evaluate the effectiveness of the proposed in-flow consistency constraints. We trained four separate models on the same dataset of unlabeled videos, each with a different combination of in-flow constraints. The latent representations of the trained models were then projected into a 2D space using PCA (Abdi & Williams, 2010) for visual analysis. The results are shown in Fig. 5. For a complementary non-linear visualization using t-SNE (Maaten & Hinton, 2008), please refer to Appendix C.

The experimental results offer compelling evidence of the importance of each constraint. (a) The combination of both directional and uniform consistency constraints results in a highly interpretable latent space. Each video's frames form a distinct, linear, and evenly spaced trajectory, effectively creating a streamline structure. This demonstrates that the model successfully captures a consistent, monotonic, and uniform progression, which is essential for learning coherent task flows. (b) Enforcing only the uniform constraint leads to a fragmented and uninterpretable representation. While the model attempts to maintain uniform progression, the absence of a directional guide causes latent vectors to meander aimlessly, resulting in a chaotic, fan-like distribution. This shows that uniform progression alone is insufficient to capture the temporal logic of a task. (c) When only the directional constraint is applied, the latent representations form a non-linear, arc-shaped structure. The model successfully enforces a monotonic direction but fails to maintain uniform progression. This non-uniform distribution manifests in the 2D space as a curved, uneven arc, which is a result of abrupt jumps and dense clustering of latent vectors. This behavior complicates the downstream extraction of a consistent task flow. (d) The baseline model, trained without any in-flow consistency constraints, produces a completely unstructured and scattered latent space. The absence of any temporal guidance results in a random cloud of points, highlighting that a standard VQVAE fails to learn a meaningful task flow representation without our proposed constraints.

**Cross-Flow Constraint Evaluation**. To verify that cross-flow consistency constraints enable the model to learn generalized task flows across demonstrations, we perform a comparative experiment between the full PFFS framework (with both in-flow and cross-flow constraints) and an ablation variant (with only in-flow, no cross-flow constraint). We visualize the latents of two distinct video demonstrations for the same task in 2D space via t-SNE. The results are presented in Fig.6.

The results offer a clear contrast between the two models. (a) The full PFFS model successfully aligns the two distinct video trajectories of the same task. The latent representations of these separate demonstrations form similar manifolds. This demonstrates that the cross-flow constraint effectively enforces a shared latent structure, ensuring that similar states across different demonstrations are mapped to a consistent location in the latent space. (b) Conversely, the model trained without the cross-flow constraint fails to align the two trajectories. While the in-flow constraint maintains the continuity of each individual trajectory, the absence of cross-flow guidance results in two inconsistent latent manifolds. This inability to align similar states highlights that in-flow consistency alone is not enough to learn a generalized task flow. The cross-flow constraint is therefore essential for creating a robust and unified representation that can guide diverse demonstrations.

### 4.3 Evaluation of Backward Planning and Forward Searching

As the core deployment components of PFFS, backward planning and forward searching work synergistically to enable policy-free task solving. We evaluate their performance across the 5 Minecraft tasks.

**Backward Planning**. We evaluate our backward planner by measuring its success rate and the length of the generated plans. For each task, we refer a success plan to finding a valid task flow from the goal state back to the initial state within 10 one-flow plan steps. The performance is summarized in Table 1.

Table 1: Backward planning performance.

| Task | Success Rate (%) | Plan Steps (Success) |
|------|------------------|----------------------|
| Harvest water | 34.5 | 4.15 ± 3.09 |
| Harvest log | 62.1 | 5.33 ± 3.00 |
| Cobblestone | 77.9 | 5.05 ± 2.73 |
| Iron ore | 46.5 | 5.03 ± 3.01 |
| Harvest sand | 33.1 | 6.72 ± 2.73 |
| Average | 50.82 | 5.26 ± 2.91 |

Our evaluation demonstrates that the backward planner effectively generates task-solving flows, though with varying success across tasks. The success rate is high for tasks like Mine cobblestone (77.9%) and Harvest log (62.1%), indicating the planner's proficiency with well-defined goals and clear state transitions. In contrast, performance is notably lower for Harvest sand (33.1%) and Harvest water (34.5%), as these tasks involve more complex and ambiguous task flows, making it difficult to capture the temporal structure of the video demonstrations. A comparison of the task flow between mine cobblestone and harvest sand is provided in Appendix A. For successful plans, the number of steps remains consistently short, with an average of approximately five steps across all tasks. This indicates that when the backward planner succeeds, it generates efficient and compact task flows.

**Forward Searching**. We evaluate how forward searching utilizes the planned task flows to solve tasks. Success is defined as reaching the goal state within a limit of exploration steps (see 3 for setup of each task). The results are summarized in Table 2.

Our forward search achieves a 26.6% average success rate, demonstrating that the task flows learned by PFFS provide effective guidance for exploration. For successful trials, the average trajectory length is 492 steps, and the average number of rollbacks is 8.13, where rollbacks refer to the frequency of failed actions where the agent returns to the previous save point in its trajectory. We pro-

Table 2: Forward Searching Performance.

| Task | Success Rate (%) | Trajectory Len. (Success) | Rollbacks (Success) |
|------|------------------|---------------------------|---------------------|
| Harvest water | 41.0 | 479 ± 244 | 9.84 |
| Harvest log | 5.0 | 356 ± 165 | 6.20 |
| Cobblestone | 57.0 | 502 ± 228 | 11.53 |
| Iron ore | 7.0 | 749 ± 73 | 4.75 |
| Harvest sand | 23.0 | 376 ± 256 | 8.35 |
| Average | 26.6 | 492 ± 193.2 | 8.13 |

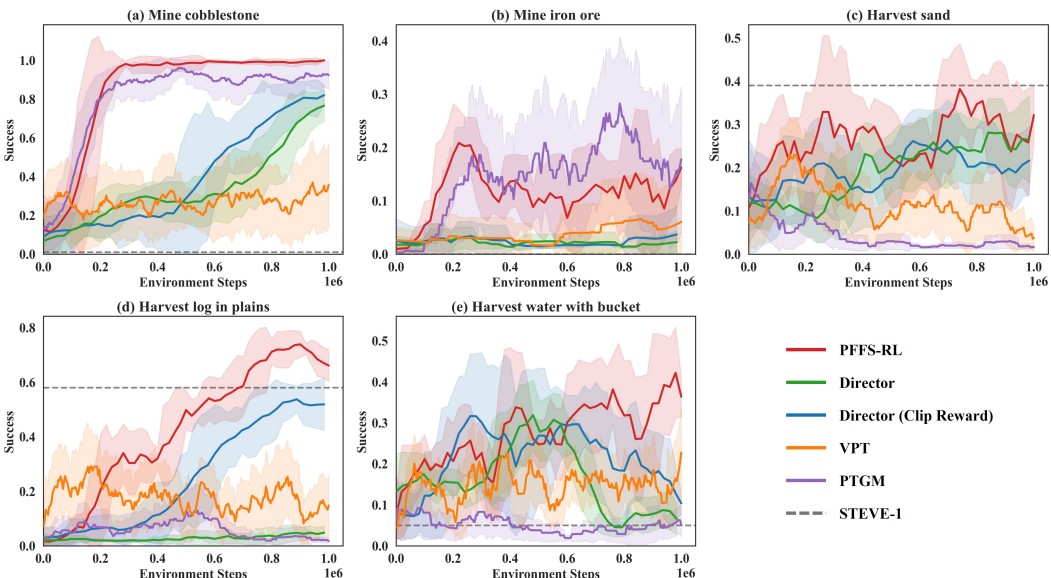

Figure 7: Performance comparison of PFFS-RL against various baselines on Minecraft.

vide two successful demonstrations in Appendix E to illustrate the exploration process. The manageable number of rollbacks and reasonable trajectory length for successful attempts suggest that our forward search is not only effective but also efficient.

### 4.4 PFFS-RL VS BASELINE RL

We compare the performance of PFFS-RL against several baselines including sota algorithms: PTGM (Yuan et al., 2024), a pre-training method utilizing goal-based models; VPT (Baker et al., 2022), a method that leverages semi-supervised pretrain on large video dataset for behavioral cloning; Steve-1 (Lifshitz et al., 2023), a multimodal model for text-to-behavior via text-video training; Director (PPO) (Schulman et al., 2017) and Director (Clip reward), which is the PPO algorithm trained with Clip-designed reward (Baker et al., 2022); The experimental results are shown in Fig.7.

PFFS-RL consistently demonstrates superior performance, outperforming all baselines on multiple Minecraft tasks. While these baselines leverage different forms of pre-training and policy guidance, they all exhibit limitations that PFFS-RL effectively overcomes which indicates that PFFS-RL's task flow guidance provides a more effective and efficient exploration signal.

### 5 CONCLUSION AND LIMITATIONS

In this work, we introduced PFFS (Policy-Free Flow Search), a novel paradigm that leverages unlabeled video demonstrations for efficient task solving. By forgoing traditional policy learning, PFFS extracts high-level, generalizable task flows from video-captured state transitions. Our framework's two-stage approach, backward planning and forward searching, provides an effective, logic-driven blueprint for solving complex, long-horizon tasks with minimal environmental exploration. We further extended this concept to PFFS-RL, demonstrating its ability to significantly boost the sample efficiency of reinforcement learning by providing task-flow-aligned rewards and guidance. Our experimental results validate that PFFS is a highly effective policy-free method for finding solutions, and PFFS-RL consistently outperforms state-of-the-art RL baselines. While our framework demonstrates strong performance, it has certain limitations. The backward planning may fail to identify a valid task flow, particularly for tasks with complex, ambiguous state dynamics. The forward searching process could also encounter challenges with complex environments, as evidenced by varying trajectory lengths and rollback rates across tasks. Finally, we acknowledge that this paper has been polished with the assistance of large language model, which may influence the final wording and style.

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

## A    TASK FLOW DEMONSTRATIONS

We visualize the task flow over video frames for two representative tasks in Fig.8.

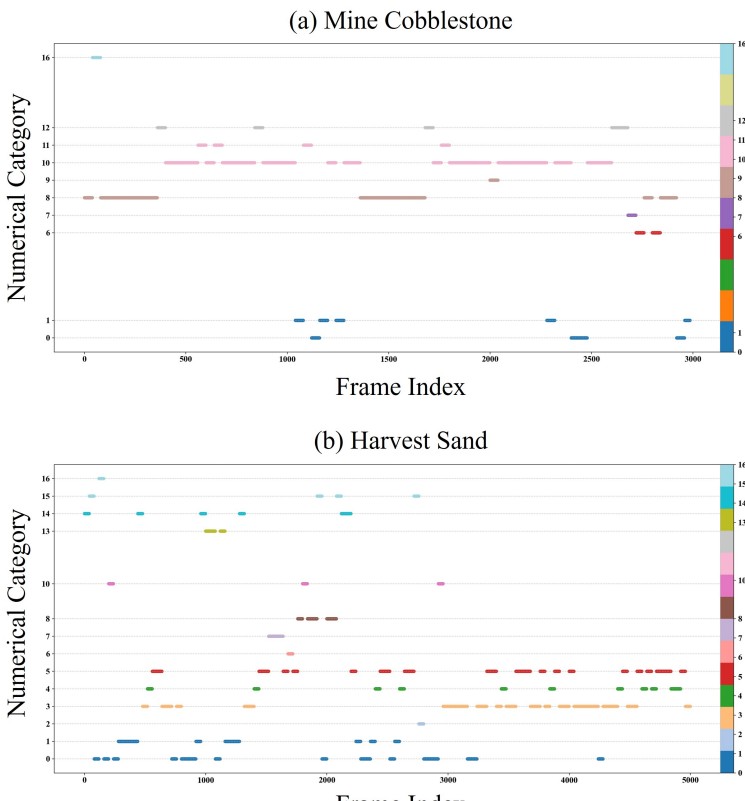

Figure 8: Task flow over video frames. (a) Mine Cobblestone displays sequential, low-variability transitions. (b) Harvest Sand exhibits fragmented, overlapping transitions, reflecting ambiguous state dynamics.

Fig.8(a): Mine Cobblestone exhibits sequential, low-variability category transitions. Each color block (numerical category) appears in long, continuous segments, indicating that PFFS effectively captures the discrete and irreversible state changes into coherent temporal latent structures, such as approaching cobblestone to mining to collecting. This well-defined task flow representation provides robust support for backward planning, which is the key reason for its high success rate.

Fig.8(b): Harvest Sand shows fragmented, overlapping, and high-variability category transitions. Multiple colors (numerical categories) interleave across frames without a dominant long-term sequence, which indicates that the task's complex and ambiguous state dynamics pose challenges for PFFS to capture a coherent temporal latent structure, including multi-stage actions. This fragmented representation directly leads to the backward planner's difficulty in modeling discrete and predictable task flows, thus explaining the lower success rate.

## B    IMPLEMENTATION DETAILS

The full OpenAI Contractor Dataset (Baker et al., 2022) contains 39M frames. We sample the raw frames at 10 FPS, resulting in 1200 complete task video clips (20-60 seconds each). For the warm-up training, we manually annotated 2% of the data (50 clips, 10 per task) with categorical labels to roughly segment task flows. These 50 clips are randomly selected to ensure their task distribution is consistent with the unlabeled data. This small annotated set is used exclusively to warm up our

VQVAE and DBSCAN, which establishes a mapping from latent representations to human-defined labels to ensure frames with the same label cluster together.

Downstream RL tasks are conducted in the MineDojo simulator (Fan et al., 2022), where the agent observes first-person 128x128 pixel RGB images and operates within a designed 12-dimensional discrete action space covering movement, camera rotation, interaction, and block breaking. During training, the agent is trained for 1 million environment steps per task. We adopt PPO for downstream RL training in our PFFS-RL framework, optimizing a weighted sum of the intrinsic rewards and a KL divergence reward to align the policy with the pre-trained task flows. The specific setup for each of the five downstream Minecraft tasks is detailed in Table 3.

Table 3: This table details the configuration for the five downstream tasks evaluated in Minecraft. For each task, the specified Language Description is used to calculate the MineCLIP reward and evaluate the Steve-1 baseline. The Initial Tools column indicates the items provided to the agent at the beginning of each episode, and Max Steps defines the maximum episode duration.

| Task | Language Description | Initial Tools | Max Steps |
|---|---|---|---|
| Harvest log in plains | "Cut a tree." | – | 1000 |
| Harvest water bucket in plains | "Find water, obtain water bucket." | bucket | 1000 |
| Harvest sand | "Obtain sand." | – | 1000 |
| Mine cobblestone | "Obtain cobblestone." | stone pickaxe | 1000 |
| Mine iron ore | "Obtain iron ore." | stone pickaxe | 2000 |

## C    T-SNE VISUALIZATION OF LATENT REPRESENTATION UNDER IN-FLOW CONSTRAINTS

To complement the PCA projections in the main paper, we provide a t-SNE visualization of the in-flow consistency constraint ablation study. As a non-linear dimensionality reduction technique, t-SNE focus on the local structure of the high-dimensional latent space. The results is shown in Fig. 9, corroborating the findings from our PCA analysis and offer a complementary view of the manifold structure.

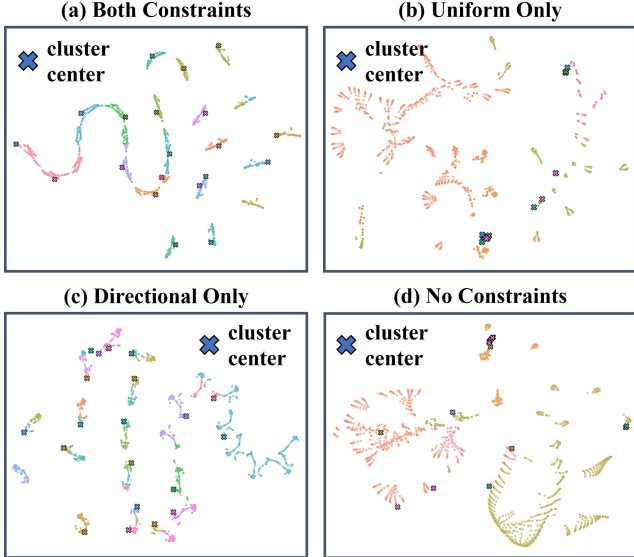

Figure 9: T-SNE visualization of the in-flow constraint ablation. Each plot corresponds to a different constraint combination, demonstrating how in-flow constraints shape the latent space.

The t-SNE visualizations provide further evidence for the effectiveness of our constraints. The combination of both directional and uniform constraints (top-left) generates a highly coherent and continuous trajectory. (a) The smooth, linear progression with evenly spaced points demonstrates the model's success in capturing both the temporal order and uniform progression of the task states. (b) In contrast, the model trained with only the directional constraint (bottom-left) creates a continuous but non-uniform arc, where points are clustered densely in some regions and sparse in others. This highlights the unique role of the uniform constraint in maintaining consistent step progression. (c) The model with only the uniform constraint (top-right) results in a fragmented and uninterpretable representation, indicating that uniform progression alone is insufficient to capture the logic of a task. (d) While the baseline model without any constraints (bottom-right) produces a completely unstructured and scattered latent space. The t-SNE plots therefore reinforce the conclusion that both directional and uniform constraints are crucial for learning a structured and interpretable latent representation.

## D

Table 4: Performance Comparison of Different Methods Across Tasks

| Method | Cobblestone | Iron | Sand | Tree | Water |
|---|---|---|---|---|---|
| PFFS-RL | **317±13** | 1733±87 | 797±48 | 493±35 | 741±36 |
| Director | 471±16 | 1968±39 | 792±36 | 968±13 | 819±31 |
| Director (Clip Reward) | 458±17 | 1913±42 | 786±41 | 823±41 | 753±29 |
| PTGM | 486±23 | 1612±135 | 953±34 | 978±15 | 981±13 |
| VPT | 980±13 | 1987±12 | 862±21 | 917±27 | 909±19 |

## E    FORWARD SEARCHING DEMONSTRATIONS

We provides a visual demonstration of the Forward Searching algorithm's operational dynamics. Fig.10 illustrate how our method guides an agent through two distinct, multi-step tasks: Harvest log in plains and Mine cobblestone. The horizontal axis represents frame samples generated during the one-flow search between adjacent categories, and the vertical axis denotes the planned task flow through backward planner, composed of sequential categories. The agent's objective is to follow this planned task flow, completing the search for each category in sequence. The figure illustrates exploration processes in two environments, from initial to goal frames. Notably, the first frame of each category is defined as a save point, if a search within a category fails, the agent returns to this save point to restart the search, ensuring efficient exploration.

In subfigure (a) Harvest log in plains, the task flow consists of 6 categories. Category 1 (red) is the initial save point, where the agent starts near a tree. As the task proceeds, the agent undertakes frame-by-frame searches to transition through categories 2–5, each corresponding to distinct stages of interacting with the tree, e.g., moving closer, positioning for logging. Category 6 (purple) is the goal category, marking the completion of log harvesting. Subfigure (b) Mine cobblestone follows a similar logic with 5 categories. Category 1 (red) is the initial save point, and Category 5 (purple) is the goal. The horizontal frame samples visualize the state transitions attempted during the search for each category, while the vertical save points anchor the start of each category's search.

These visualizations demonstrate the algorithm's capability to effectively model complex task flows and navigate an agent through them by leveraging the iterative One-Flow Search.

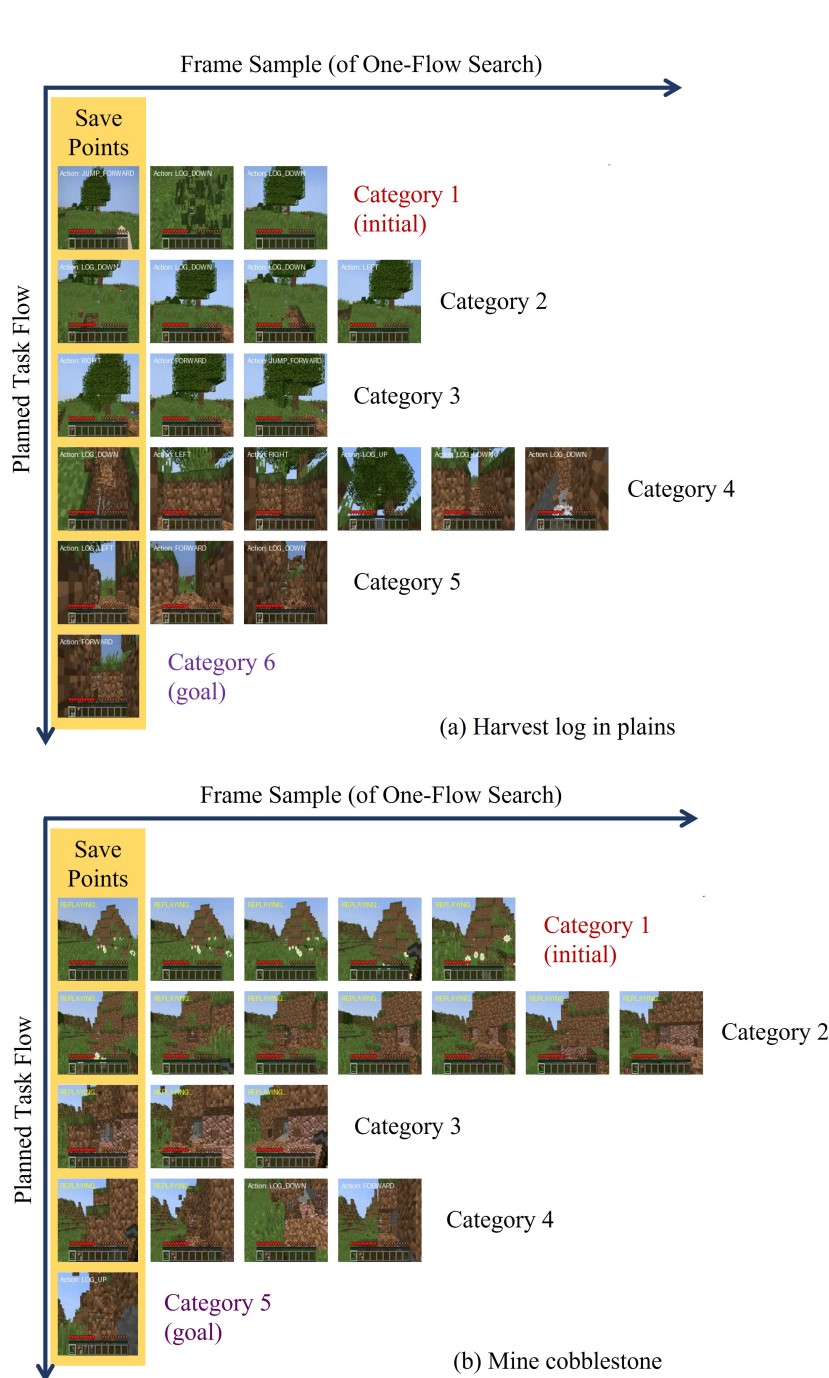

Figure 10: Visualization of One-Flow Search in two Minecraft tasks. Both tasks feature save points (first frame of each category) enabling rollback-based exploration, with horizontal frame samples showing state transitions during inter-category search. (a) Harvest log in plains: the planned task flow includes 6 sequential categories. (b) Mine cobblestone: the planned task flow includes 5 sequential categories.

