# OpenReview forum: "Beyond Policy Training: Solution Search via Task Flow Learning and Planning from Unlabeled Videos"
_ICLR.cc/2026/Conference — ICLR 2026 Conference Withdrawn Submission_

### Official Review · Reviewer_p62V · 2025-10-26

**Soundness:** 2
**Presentation:** 2
**Contribution:** 2
**Rating:** 4
**Confidence:** 2

**Summary:**

This paper introduces Policy-Free Flow Search (PFFS), which models task stage transitions autoregressively to form a coherent task flow. The proposed in-flow constraint helps make meaningful latent representations within a trajectory and cross-flow constraint helps make different demos of the same task to generate alike latent distributions. PFFS performs backward planning to generate a goal-to-initial task flow, then executes forward search to solve the task along this flow. The author also extends PFFS to PFFS-RL with PPO, it shows fast convergence and higher performance on almost all 5 selected Minecraft tasks.

**Strengths:**

- Backward planning is interesting and this helps following forward searching to be easy and fast.
- In-flow constraint helps make meaningful latent representations within a trajectory. Cross-flow constraint helps make different demos of the same task to generate alike latent distributions.
- PFFS-RL shows fast convergence and higher performance on almost all 5 selected Minecraft tasks.

**Weaknesses:**

- The experiment is inadequate, solely Minecraft is tested. The author may consider adding more experiments on OGBench or D4RL.
- Figure 2 only depicts a single video which brings in-flow loss. But without multiple videos here, it is difficult to understand the cross-flow loss through this picture.
- Figure 2 shows 3 clusters after DBSCAN, but there are 5 clusters in the task flow distribution part, which does not match.

**Questions:**

- In Section 3.3, "If the agent enters a category that deviates from the plan or reaches the maximum search step length, it returns to the latest save point or $s_{init}$ if no save points exist." I think this works because it is in a simulator, but how to apply this method to the real world? It's very difficult to reset the environment even with a human in the loop.
- What's the detailed difference between backward planning and forward planning? In my perspective, they are both one-flow plan/search, by first ignoring self-class and then doing a probability-based sampling.
- How to adjust the parameters of DBSCAN, $\epsilon$ and **minPts**? Is PFFS robust to these hyper-parameters?

---

### Official Review · Reviewer_BBWu · 2025-10-29

**Soundness:** 2
**Presentation:** 3
**Contribution:** 2
**Rating:** 4
**Confidence:** 3

**Summary:**

The paper learns a latent space that can encode task flow—the order in which the task should be executed—from action-free videos and uses it to guide exploration. The latent is trained with two self-supervised objectives: (1) in-flow consistency, making adjacent states advance monotonically and uniformly toward the goal within each demo; and (2) cross-flow consistency, which aligns the ordering of latent categories (i.e., clusters) across demonstrations of the same task. The resulting representation serves as a roadmap: a backward pass samples feasible intermediate states from the goal back to the initial state, and a forward pass steers the agent toward these intermediates. In the Minecraft simulator, the learned representation structures trajectories smoothly from initial states to the goal and enables more sample-efficient RL than baselines.

**Strengths:**

- The introduction and Sec 2.2 provide a clear primer on RL exploration and convincingly motivate policy/action-free exploration.
- The paper offers thorough analyses: qualitative visualizations of the learned representation, ablations of both objectives, and quantitative results on downstream RL.
- It consistently outperforms action-based methods (e.g., VPT) across all five Minecraft tasks, highlighting the efficiency of action-free task representations.

**Weaknesses:**

1. What helps: save points vs. representation?
- The method restarts episodes from “save points, which isn’t practical in real-world RL. This also confounds attribution: are the gains due to the reset strategy or the learned representation? Please quantify the reset’s contribution to sample efficiency—e.g., report results without resets—and clarify the source of improvements. If I’ve misunderstood the method, please correct me.

2. Action-free video baselines
- Prior works have used action-free video for exploration, but it is not discussed. Please position your method relative to the literature (e.g., [1, 2]) and—if appropriate—include a brief empirical comparison; this will strengthen the paper.

#### References
- [1] Escontrela et al., "Video Prediction Models as Rewards for Reinforcement Learning", NeurIPS 2023
- [2] Ma et al., "Video Prediction Models as Rewards for Reinforcement Learning", ICLR 2023

**Questions:**

- Q1. How many videos were used to train the latent, and how does performance scale with dataset size? Can your method efficiently leverage larger, more diverse (multi-modal) task flows—i.e., different ways of solving the task? A sensitivity/scaling analysis would be helpful.
- Q2. Lines 311–312 mention 50 annotated clips. How critical are these labels? Would performance fail or degrade notably without them?
- Q3. For VPT, did you only use a large-scale pretrained checkpoint to label actions on your data, or did you also fine-tune on your dataset with action labels? If former, is the comparison fair given your additional annotations?
- Q4. Do Figures 5 and 6 depict in-domain (used to train the latent) trajectories or out-of-domain data? If they’re in-domain, how would the latent visualize on OOD inputs?

---

### Official Review · Reviewer_QhGW · 2025-10-30

**Soundness:** 3
**Presentation:** 2
**Contribution:** 2
**Rating:** 2
**Confidence:** 3

**Summary:**

The paper studies online exploration for reinforcement learning from unlabeled prior video dataset. The authors propose to learn a structured latent representation of the video observation where the task progression corresponds to a steady, uniformly spaced latent vector from the initial state to the goal state. This representation space allows the states to be clustered into discrete categories, which then allows a backward planner to model the reverse transition probability for the clusters. With sample-based planning, the backward planner can generate a sequence of clusters from the goal state to the initial state and then guide the RL exploration using such generated plan (e.g., positive reward when the agent follows the plan and negative reward when it is off-track). The proposed method is evaluated on the Minecraft benchmark where it outperforms prior baselines that do not leverage such planning.

**Strengths:**

- The paper tackles online exploration from unlabeled prior video, which is a very challenging and important research direction.

- The paper is easy to read and the proposed method is well-motivated and described in great clarity.

**Weaknesses:**

- "Current methods either rely on handcrafted prior cues or leverage annotated data to learn exploration guidance, which still suffers from annotation cost": this is not true. There are many prior methods that do not use annotations (e.g., [1-4])

- Empirical results are a bit weak considering the complexity of the proposed method. In addition, I find the baselines to be a bit limited. How does the proposed method compare to standard model-based RL approaches? (e.g., DreamerV3 [10]) What about representation learning approaches (e.g., ICVF as also cited by the authors).

- Table 2 reports the success rate of the forward searching performance. The success rate looks great but it tells little without a proper comparison. How does it compare to random exploration without planning? What about PPO with exploration reward bonus?

- Discussions of important related work are missing. To list a few lines of work that I am more familiar with,
  - Curriculum learning [5]: much of the motivation of curriculum learning is similar to the motivation of the proposed method. Typical curriculum learning methods propose tasks with intermediate difficulties and learn to tackle them first before moving on to the hardest/target task that needs to be solved. Among them, [6] explicitly uses backward reachability curriculum similar to the proposed backward planning to guide learning.
  - Model-based RL [7]: planning and search is commonly used in model-based reinforcement learning. Among these, backward planning (which is directly related to the proposed method) has also been explored  extensively (e.g., [8-9]).

[1] Li, Qiyang, et al. "Accelerating exploration with unlabeled prior data." NeurIPS 2023

[2] Hu, Hao, et al. "Unsupervised behavior extraction via random intent priors." NeurIPS 2023

[3] Wilcoxson, Max, et al. "Leveraging skills from unlabeled prior data for efficient online exploration." ICML 2025

[4] Kim, Junsu, Seohong Park, and Sergey Levine. "Unsupervised-to-online reinforcement learning." arXiv preprint arXiv:2408.14785 (2024).

[5] Narvekar, Sanmit, et al. "Curriculum learning for reinforcement learning domains: A framework and survey." Journal of Machine Learning Research 21.181 (2020): 1-50.

[6] Ivanovic, Boris, et al. "Barc: Backward reachability curriculum for robotic reinforcement learning." 2019 International Conference on Robotics and Automation (ICRA). IEEE, 2019.

[7] Moerland, Thomas M., et al. "Model-based reinforcement learning: A survey." Foundations and Trends® in Machine Learning 16.1 (2023): 1-118.

[8] Goyal, Anirudh, et al. "Recall traces: Backtracking models for efficient reinforcement learning." arXiv preprint arXiv:1804.00379 (2018).

[9] Edwards, Ashley D., Laura Downs, and James C. Davidson. "Forward-backward reinforcement learning." arXiv preprint arXiv:1803.10227 (2018).

[10] Hafner, Danijar, et al. "Mastering diverse domains through world models." arXiv preprint arXiv:2301.04104 (2023).

**Questions:**

N/A. Please see the questions above on the baselines (e.g., model-based methods, representation learning methods, basic exploration approaches).

---

### Official Review · Reviewer_fsLQ · 2025-10-31

**Soundness:** 2
**Presentation:** 4
**Contribution:** 3
**Rating:** 6
**Confidence:** 5

**Summary:**

The paper proposes Policy-Free Flow Search (PFFS), a framework for solving tasks directly from unlabeled videos without relying on policy learning. It learns structured task flows using two self-supervised objectives, in-flow (temporal consistency within each video) and cross-flow (structural alignment across videos), to capture generalizable task progressions. At test time, the model performs backward planning from the goal to the start, and then a forward search along this flow to reach the goal with minimal exploration. The authors also extend this idea to PFFS-RL, which integrates task-flow–aligned rewards into reinforcement learning. Experiments on Minecraft show that PFFS can find task solutions with little exploration, and that PFFS-RL improves over several RL baselines.

**Strengths:**

1. The idea is novel and well-motivated. Using unlabeled videos to extract task structure without any explicit policy is genuinely new and quite intuitive.

2. The core idea of backward planning + forward search from unlabeled demonstrations is intuitive and well presented.

3. The paper is well organized and easy to follow. Figures and diagrams do a good job illustrating the concepts.

4. The consistency losses are clearly motivated, and ablation studies show their contribution.

5. Experiments demonstrate clear gains in exploration efficiency

**Weaknesses:**

1. The paper introduces multiple loss components but doesn’t discuss how sensitive the results are to their weights or whether they ever conflict.

2. The save-point sampling seems crucial to the results, yet the paper doesn’t analyze what happens without it.

3. While labeled as “policy-free,” the method still depends on a small labeled subset for warm-up, which weakens that claim.

4. Forward search success rates are fairly low, and it’s unclear how this would scale to more complex or noisy environments.

5. The clustering stage (DBSCAN) is an important part of the pipeline, but there’s no analysis of its robustness to parameters or data noise.

**Questions:**

1. How stable is the framework with respect to different loss weights or cluster counts?

2. Would PFFS still perform reasonably without the save-point sampling strategy?

3. Can the learned flows transfer across tasks, or is re-training needed each time?

4. How would the system handle stochastic or partially observable transitions?

5. How stable is DBSCAN across different video datasets or varying cluster counts?

---

### Note · Authors · 2025-12-04

I have read and agree with the venue's withdrawal policy on behalf of myself and my co-authors.